# Corporate Social Responsibility and Social Capital: Journey of Community Engagement toward Community Empowerment Program in Developing Country

**Bambang Rudito ***, **Melia Famiola and Prameshwara Anggahegari**

School of Business and Management—Institut Teknologi Bandung, Bandung 40132, Indonesia
* Correspondence: brudito@sbm-itb.ac.id

**Abstract:** The importance of social capital in analyzing economic progress and development has long been acknowledged. However, little research has actually looked into how social capital functions and what role it plays in community development initiatives. This case study explores communities' experiences of receiving CSR funds informing a community development program. The informants of this study were the participants of the CSR program from three communities forested by three companies' CSR, which received reward funding from the Indonesian government. By using qualitative data collection, this study interviewed five to seven people who participated in the CSR programs or approximately 16 people. This study found five ways that social capital contributes to the success of a CSR program. They are (1) increasing the likelihood that people will participate in the program; (2) lessening individual ambiguity caused by certain program implementation uncertainties; (3) fostering morale and motivation among participants; (4) promoting knowledge sharing; and (5) strengthening the sense of togetherness. In the future, additional case studies from various regions of Indonesia and other nations may examine local conditions and diversity when implementing CSR programs.

**Keywords:** social capital; community engagement; community-based program

## 1. Introduction

Corporate Social Responsibility (CSR) has emerged as a major topic for debate among business professionals, academics, and governments. Now, the public is paying attention as well, especially businesses operating in developing nations. The public has high hopes for business to significantly enhance their social lives. Therefore, many businesses in developing nations work on a variety of social initiatives, including health care, education, economic welfare, infrastructure development, and environmental protection, in order to improve their reputation and gain legitimacy in the region where they operate [1,2]. It is not an easy procedure, nevertheless, to build a project in a community. The program's success is impacted by several elements, including the outcomes if the program is implemented in an area with a character quite dissimilar from the corporate culture. Multinational enterprises, such as those engaged in the mining, oil, and natural resource industries, frequently discover this issue while operating even in remote locations.

The purpose of the company's social activities is not discussed in this article or evaluated. Instead, we examine how corporate initiatives through CSR programs and their attempts to improve community wellbeing and resilience could achieve their missions. Today, many companies have adopted a share value approach [3] that expect CSR could provide the benefit of social change to the targeted community and the corporation simultaneously. In other words, CSR is one of a company's investment models to sharpen its role in society [4]. Therefore, capital spending should be thoughtfully planned to gain a positive impact.

Social change must be understood as a process [5]. It occurs due to the actions and interactions of individuals influencing a change. Of course, many institutional factors affect an individual's negotiation for change. However, the dynamic of social interaction in the process has more significant implications for an individual who works for change [5]. Therefore, this study assumes that social capital within a community will closely influence the step-by-step process of how real change happens. Therefore, paying strict attention to the social capital in the CSR's targeted community will make a company's CSR achieve its ultimate goals.

Social capital is a theory that examines how people are connected, how they build trust, and how willing they are to abide by the standards that become their consensus [6]. This strategy was widely used by academics, decision-makers, and practitioners as a helpful idea relating to community development [7,8]. In order to understand how social capital helps in the process and the community's involvement with the program. This study has a research question: what is the role of social capital within the process of introducing and implementing the program? Although the phrase "social capital" originates from research relating to communities, there has been little discussion of the function of social capital before, during, and after an empowerment program.

This research is intended to add to both theoretical and practical advancement by discussing the highlighted problem. In terms of theory, this article offers an alternative viewpoint on the function of social capital, how it operates, and what qualities of its function determine the effectiveness and success of a community's engagement. On the other hand, from a practical standpoint, we envisage that it can provide insights to every stakeholder who intends to carry out a community-based program, what social aspect they need to consider before running the program, and how they can use the social capital of the targeted community to accelerate and ensure the program run as was planned.

Therefore, the following is the outline for this paper. The literature review on social capital will be included in the second section, along with findings from earlier studies on its significance in comprehending social phenomena. The research methodology is explained in the following chapter, which is followed by the findings, a discussion, and a conclusion.

## 2. Literature Review

### 2.1. CSR and Social Change

The CSR program will typically take the shape of community empowerment in the form of community development, which contributes to the positive transformation of communities and social change [9]. Particularly for CSR initiatives in developing countries, the community development-focused CSR model is regarded as the best alternative to purely charitable initiatives. Community development is described by Philips and Pittman [10] as an educational process that enables members of the community to discuss issues and jointly create solutions. The process involves making an effort to improve a certain component of community life, which strengthens the community's institutional and interpersonal relationship structure. In other words, community development focuses on the procedure of training individuals on how to collaborate with one another to address their shared challenges [2].

Any community development program should aim to bring about good social change, which is typically focused on enhancing well-being and addressing localized poverty [11,12]. The goal of building a more resilient community is currently a key concern in CSR implementation [13,14]. One of the main topics that businesses are being requested to focus on in order to carry out their CSR programs is climate change and other environmental difficulties [15]. It will be in the company's best advantage to use CSR to help solve the clean water and sanitation issues faced by drought-stricken and low-income communities as an illustration of how climate change is having a significant impact on agricultural areas [16]), particularly in light of the requirement for enterprises to support sustainable development and report on their progress toward the Sustainable Development Goals (SDGs) [17].

The program will be received differently by various communities because social change is delinear [11]. This disparity in response is brought on by a variety of factors, including: (1) an understanding of the level of urgency of the problems and issues chosen from the perspective of the community that is the target of CSR; (2) internalization of program values introduced to the community through socialization and communication of the CSR agenda; and (3) other technical issues with the implementation of the coaching program related to the conditions that exist in the affected community [18]. In other words, providing social engineering is required in order for the CSR agenda to have a good societal impact. In this study, we attempt to determine how social capital functions to support the level of community engagement and acceptance for the implementation of CSR programs targeted at them, particularly for businesses that engage in CSR activities in the neighborhood with the goal of fostering positive social change through empowerment, which is the hallmark of CSR practices in developing countries.

### 2.2. Social Capital

The idea of "social capital" was first established as a sociological topic in the latter part of the nineteenth century. This phrase was developed to provide a logical framework for comprehending human behavior and how society influences people's perceptions and motivations [19–21]. This phrase emphasizes the contribution that interpersonal relationships and social engagement contribute to the growth of an individual. Social capital is viewed as an asset that is ingrained in relationships between people, communities, and societies [20,22,23]. In light of this, the term "capital" is mixed with the word "social" [24] and adapted from the well-established ideas of financial and human capital [19]).

Social capital is described as "networks together with shared norms, attitudes and understandings that allow co-operation inside and among groups" by the World Bank and the Organisation for Economic Co-operation and Development (OECD). Social capital, defined by The World Bank (1999) as the institutions, relationships, and norms that explain the type and volume of social interaction within a society, serves as the society's binding agent [8,20].

There are two distinct mainstream definitions of social capital used by researchers in its development [19,23,25]. First, social capital is a characteristic of an individual [26,27]. In this case, social capital is viewed as an interpersonal competency that a person possesses in both formal and casual interactions with other people. Coleman (1988) advocated three social capital figures which may help some individual acts when they are in a structure: (1) Obligation and expectation; this social capital is formed as conformity of people's trustworthiness in a structure; (2) the ability of information flow inside the structure that might support an action; and (3) the present norm that becomes a guideline for every individual action within the structure.

Social capital is also regarded as a community attribute. Social capital is defined here as a model of interaction between individuals who participate in a structure or a community [8,24,28]. According to Fukuyama [8], social capital is the networking that occurs and the way that trust grows within a community or a group. Three aspects of social capital, according to Yokoyama and Ishida [8]: (1) Networks and memberships; a community's network of structures is described [8,23]. A strong network among people in a structure or arena can lower social risk and transaction costs. It can also foster social trust, collective action, and reciprocity [20,29].

### 2.3. Role of Social Capital

Social capital is a crucial component of linking between and among people as a result of a social system. Social capital joins ownership among several participants in a relationship, differing from other forms of capital. No single player has or is eligible to hold the ownership rights. Even social capital is important when used, but trading it is difficult [6,23]. For instance, friendship exists amongst people and does not belong to a single individual. This type of relationship's ownership could not be transferred to others.

The importance of social capital has been noted by academics. First, North [30] describes it as a cost decrease for certain tasks or "allocative efficiency". Achieving a goal without incurring high costs is possible when individuals within a system have healthy social relationships with one another [22]. For instance, Nahapiet and Ghoshal [23] show how efficient communication among participants in a structure may reduce redundancy and increase information dissemination efficiency. Putnam [28] also found that social capital in the form of a high level of trust might lower the likelihood of opportunism and reduce monitoring expenses, which could lower the cost of a transaction.

Second, social capital might boost creativity, improve adaptability, and speed up the learning process. An organization's social capital may foster cooperative behavior that fosters innovation and creativity, according to prior research [23,31]. Additionally, social capital may make it easier for people to develop their abilities [26]. Coleman [26] provided an example of social capital in a family. Children's interactions with their parents may provide them with opportunities to learn a lot about responsibility. As another illustration, Maskell [21] demonstrated how social capital might lower the cost of transactions and teamwork during the production of innovations.

Scholars have also noted that social capital may not always provide a favorable outcome. According to Coleman [26], social capital may do harm to others through its use. For instance, some strong social norms may encourage cross-group cooperations that result in effective group performance. On the other hand, this circumstance has the potential to limit creativity, innovation, and informational openness.

This study aims to provide more specific information about the function of social capital in a community's engagement with an empowerment program run through CSR programs. We do not only pay attention to one aspect of social capital. We think of social capital as having two functions and two dimensions. We contend that since each topic covered in this literature review is interconnected, it is impossible for them to be distinguished in actual usage.

## 3. Materials and Methods

### 3.1. Data Collection

This study is qualitative with a purposive sampling approach [32–34]. We interviewed representatives of the community who are directly involved in three CSR programs that have successfully guided the community to improve their well-being [35]. The three CSR programs have brought their company to receive an award from the Indonesian government as one of the companies implementing CSR with a broad impact.

We work with five to seven people as our informants from each CSR program. These people are beneficial individuals in the programs. They make repeated observations before being invited to participate in interviews pertinent to the study's goals [36,37].

### 3.2. Case CSR Sampling

In this study, three CSR cases are used. The cased CSRs in this study are the CSR programs that many stakeholders, including the Indonesian government, recognize. The CSR programs are in the form of community development and have been running for more than five years. The following Table 1 is a profile of the CSR program, the company, and the number of beneficial individuals of each CSR program that has become an informant of the study.

From Biofarma's CSR initiatives, the first case was gathered. Biofarma is a biotechnology business based in Indonesia. It is one of the state-owned businesses in the drugstore. For the execution of its CSR program, Biofarma is one of the businesses with operations in Indonesia that was given the highest PROPER award by the Indonesian government.

The breeding or preservation of the purebred Garut sheep, which possesses unique genetic traits, is one of Biofarma's signature CSR initiatives. The horns and weights of Garut sheep are rounded and large. Due to unchecked farming practices, such as crossbreeding,

where the sheep are mixed with other types of sheep only to satisfy market demands, it is believed that the population of these sheep has begun to decline.

**Table 1.** Case studies sample.

| Case Study | Program | Number of Key Informants |
|---|---|---|
| Biofarma's CSR | Application of biotechnology that can be utilized by the community for the purification of the Garut sheep breed | 5 |
| Star Energy's CSR | Empower villagers in the surroundings of Star Energy operation through training to create products from local main product milk and entrepreneurship | 7 |
| PGE's CSR | Empower local villagers in entrepreneurship to innovate mushroom products and traditional snacks | 5 |

Through this program, Biofarma introduces the application of biotechnology that can be utilized by the community for the purification of the Garut sheep breed. This program is carried out on sheep breeders in Wanajaya Village, Kec. Wanarja, Kab. Garut, West Java. The activities carried out are assistance in the form of management training, animal health counseling, and so on in the process of cultivating Garut sheep, which should be carried out by sheep breeders.

The specific case came from a CSR initiative run by PT. Star Energy. Located in Pangalengan, South Bandung, West Java, Star Energy is one of the private businesses with a concentration on geothermal energy exploration. In the vicinity of its operation region, Star Energy conducts community projects. Our team has been a part of this initiative since its inception, and we are currently in the period of establishment. Through this program, the Indonesian government awarded Star Energy Gold PROPER status. The highest government recognition for an organization's CSR program is this award. The program will train the villagers in the production of milk, their main product, and assist them in looking for possible markets. We observe and speak with about seven of the program's communities.

The third case was from the CSR program of PT. PERTAMINA Geothermal Energy (PGE). PT PGE is a state-owned company in the oil and gas industry. The location of this project is in Kamojang, North Bandung, West Java. PGE also received a gold PROPER award from the Indonesian government. The CSR program observed for this study was capacity building for mushroom farms and training for making a traditional local snack for villagers around its operation area. The program also helps its participants to access their potential market.

## 4. Findings and Discussion

The findings of our conversations and interviews indicate, in general, that social capital in a community plays a crucial part in fostering community participation with CSR initiatives targeted at them. Throughout the CSR program process, social capital is engaged at three different levels. We refer to it as community engagement in the CSR journey. The trip has three phases: engagement, involvement, and awareness. Figure 1 illustrates the role of social capital at each level of the community involvement pathway.

The degree of intention to participate in the CSR agenda is first determined by awareness. The deep interaction between program informants and potential participants offers a fantastic chance to pique interest in the CSR agenda among potential participants.

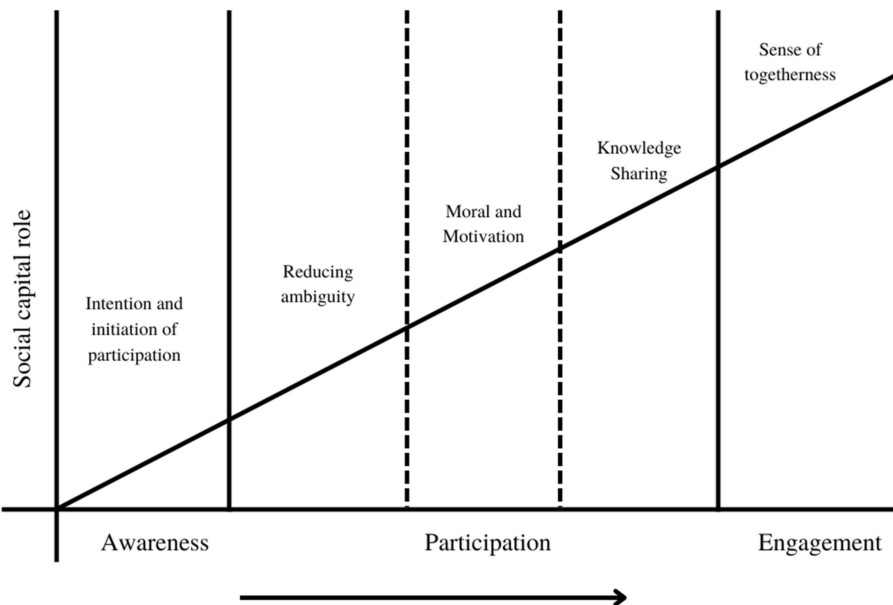

**Figure 1.** Role of Social Capital to Community Journey of Engagement in CSR Programs.

In other words, social capital is crucial in promoting community members' first commitment. The influence of peer invitation on an individual's decision to participate in a new program is well known. Peer invitation might be a successful strategy to influence numerous participants to join the program, even while the informant does not seem to fully appreciate the benefits of the program. The lines that follow demonstrate how influential peer invitation is encouraging participation among the villagers or group.

"Saya diberitahu dan diajak pegawai kelurahan, yang kebetulan tetangga saya juga" (Biofarma's CSR_informant 1)

"I was notified and invited by a village employee, who happens to be my neighbor too" (Biofarma's CSR_informantt 1)

*"Masalah tentang domba garut memang sudah menjadi diskusi kami di group peternak domba disini, jadi ketika ada teman yang memberitahu dan mengajak ikut, saya ikutan juga"* (Biofarma's CSR_informant 5)

"The issue of sheep has indeed become our discussion in the sheep breeder group here, so when a friend tells me and invites me to come along, I'm in" (Biofarma's CSR_informant 5)

*"Banyak teman-teman kelompok PKK kami mengikuti kegiatan ini, jika banyak orang yang terlibat, kami tidak akan merasa kesepian"* (CSR-Informant Star energy 4)

"Many friends of our PKK group join these activities, if there are many people involved, we will not feel lonely" (Star energy's CSR-Informant 4)

*"Jujur saya awalnya tidak tahu tujuan program ini, saya hanya datang atas undangan teman-teman"* (CSR-Informant Star energy 3)

"Honestly, I did not know the purpose of this program initially, I just came by invitation of my friends" (Star energy's CSR-Informant 3)

*"Saya mengikuti program ini karena undangan kerabat saya"* (PGE CSR-Informant 1)

"I join this program cause of the invitation of my relative" (PGE CSR-Informant 1)

*"Saya ikut karena undangan teman-teman yang lain di kelompok usaha kecil saya, walaupun saya tidak mengerti program apa itu, saya ikut saja"* (PGE CSR-Informant 2)

*"I participated because invitation of other friends in my small business group, although I do not understand what kind of program is it, I just participate"* (PGE CSR-Informant 2)

Second, in the stage of involvement, we contend that CSR capital plays thirty crucial roles at this point. Getting rid of uncertainties comes first. Since the majority of program participants do not precisely comprehend the program's goal, this is a problem. They only enroll in the program at the invitation of their peers. Because of this, there is frequently some uncertainty caused when programs are implemented.

We have seen that some participants have even accepted invitations from their peers. They contribute part of their enthusiasm for the presentation. As a result, when the program falls short of their initial expectations, some confusion may arise, leaving some of them unsure of whether to keep participating in the program or not. This uncertainty and purpose of terminating might be lessened by the tight relationships among their peers. The program's pleased participants may assist the program's dissatisfied participants in reducing their uncertainty and attempting to align their expectations and motivation with the program.

> *"Sejujurnya saya awalnya mengira teknologi yang diperkenalkan itu sulit, tapi karena kita sudah lama mengenal konselor, jadi kalau bingung kita tanya ke mereka"* (Bio-farma's CSR_informantt2)

> "To be honest, I initially thought the technology introduced is difficult, but because we had known the counsellors for so long, so if we are confused, we ask them" (Bio-farma's CSR_informant 2)

> *"Yang saya pahami program ini adalah mendukung agenda pemerintah. Artinya bagus makanya saya ikut, karena selain menambah ilmu, pesertanya kebanyakan orang yang saya kenal cukup baik, jadi kita tidak akan bingung atau canggung"* (Biofarma's CSR _informant5)

> "What I understand the program is support the government agenda. It means good so that is why I participate, because, in addition to adding our knowledge, the participant is most people whom I know quite well, so we would not feel confused or awkward" (Biofarma's CSR_informant 5)

> *"Kalau ada teman, saudara atau saudara yang ikutan, kami tidak bingung"* (CSR-Informant2 Star Energy2)

> "If there are friends, siblings or relatives who participated in a, we are not confused" (Star Energy's CSR-Informant 2)

> *"Awalnya saya diajak oleh kakak saya untuk mengikuti program ini. Saya masih tidak mengerti, tetapi karena ada saudara saya, saya bisa bertanya padanya. Setelah saya berpartisipasi, itu cukup bisa dimengerti. Ternyata bermanfaat."* (CSR-Informant Star Energy 7)

> "At first, I was invited by my brother to participate in this program. I still did not understand, but because there is my brother, I can ask him. After I participated, it is quite understandable. It turns out to be useful." (Star Energy's CSR-Informant 7)

> *"Kalau kita ikut program dan ada orang yang kita kenal, alangkah baiknya, jadi kalau tidak paham, kita bisa saling membantu"* (PGE CSR-Informant 2)

> "If we join in the program and there are people that we know, it would be great, so if you cannot understand, we can help each other" (PGE CSR-Informant 2)

> *"Menurut ketua kelompok kami, programnya untuk meningkatkan perekonomian kami, jadi saya ikut"* (PGE CSR-Informant 2)

> "According to the chairman of our group, the program is to improve our economy, so I participate" (PGE CSR-Informant 2)

Along with removing uncertainty, the social capital and intimate connections within the targeted group will inspire and motivate individuals. Software frequently experiences ups and downs. Due to personal reasons, one member leaves and wishes to terminate. For

instance, in the CS program run by Star Energy, the community leader had to leave the program after giving birth to her child. When those individuals referred to their motivation and initial aim, the program resumed activity after going dormant for a while. To restart the program, they spritely hunt for one another.

We demonstrate using our observations that this moral may result from a deep emotional connection and relationship. Each person may draw a connection between their initial expectations of the program and the group's support for reworking to accomplish their shared purpose. Only through engaging in frequent communication among the members could a strong emotional bond be formed. Regarding the topic covered in the quotes below, our respondent provided a supportive comment.

> *"Kebetulan teman-teman di grup saya adalah orang-orang yang dekat dengan saya, jadi kita bisa saling mengingatkan dan memotivasi"* (Biofarma CSR_informant 3)

> "Coincidentally my friends in my group are people who are close to me, so we can remind and motivate each other" (Biofarma's CSR_informant 3)

> *"Kalau kita lakukan bersama-sama, maka ketika kita merasa kesulitan, kita bisa saling membantu"* (Biofarma's CSR_informant 5)

> "If we are doing it together, so when we feel some difficulties, we can help each other" (Biofarma's CSR_informant 5)

> *"Program ini hampir dihentikan karena leader kami melahirkan, jadi dia tidak bisa berpartisipasi secara intensif. Untungnya, kami di sini sudah dianggap sebagai keluarga, jadi kami bisa saling memotivasi lagi. Kami menyadari bahwa program ini membawa kebaikan bagi kami."* (CSR-Informant Star energy 1)

> "This program was almost terminated because our leader gave birth, so she cannot participate intensively. Luckily, we were regard we here as a family here, so we can motivate each other again. We realised that the program bring goodness for us." (Star energy's CSR-Informant 1)

> *"Berjuang dengan beberapa masalah akan lebih mudah diselesaikan ketika kita tidak sendirian"* (CSR-Informant Star energy 3)

> "Struggling with some problem will easier to solve when we are not alone" (Star energy's CSR-Informant 3)

> *"Memulai yang baru sendiri itu sulit, tetapi melakukannya bersama-sama membuatnya mudah. Seperti kata orang-orang kami ini gotong royong (kerjasama kolektif)"* (PGE CSR-Informant 2)

> "Starting a new one on your own is hard, but doing it together make it easy. Like our people said this a gotong royong (collective collaboration)" (PGE CSR-Informant 2)

The third type of social capital is knowledge sharing, which is present in the community as a result of the closeness of inter-individual interactions in the company's CSR efforts. Participants in this case study may share information more easily if they are socially connected. It is difficult for it to change to a new technique or software. The program's goals might be accomplished by participants exchanging experiences and expertise. Sharing of knowledge inside this community or group might be regarded as a collecting action. Our research showed that the development of this sharing environment requires trust among participants in addition to relationships among program participants.

For instance, the observation demonstrates that sharing often occurs among people who get along well and connect with many areas of their daily lives. In other words, it requires a psychological bond between individuals to promote information exchange. The refusal to provide the information is frequently met with resistance from those who are dealing with external issues or because of unpleasant prior interpersonal interactions. Prejudice acts as a psychological barrier that prevents knowledge from being shared. We saw two program participants who had long-standing family disputes during our

observation. Although both parties claimed the disputes were resolved, their relationship and relationships with other participants were not. They were forbidden from sharing one another. Therefore, they usually found other partners to do so.

> *"Program pelatihan dan penyuluhan ini memperkenalkan hal-hal baru bagi kami, tidak mudah untuk dipelajari, tetapi dengan belajar bersama, mempermudah, jika saya tidak mengerti, saya bisa bertanya kepada beberapa teman di kelompok saya."* (CSR_informant Biofarma 4)

> "The training and consultation program is introducing new things for us, it is not easy to learn, but by learning together, make easier, if I don't understand, I could ask some friends in my group." (CSR-informant Biofarma 4)

> *"The farming methods introduced is a new thing for me, sometimes it is difficult, but I do not take it hard, if I do not understand, I usually ask a friend in my group when we practice."* (Biofarma's CSR_informant 5)

> "Metode bertani yang diperkenalkan merupakan hal baru bagi saya, terkadang sulit, tapi saya tidak ambil berat, jika saya tidak mengerti, saya biasanya bertanya kepada teman di kelompok saya ketika kami berlatih." (CSR_informant Biofarma 5)

> *"Saya mengikuti pelatihan membuat yoghurt; pelatih tidak bisa berbahasa sunda. Jika Anda menggunakan bahasa Sunda, akan jauh lebih baik untuk dipahami. Untung ada beberapa teman yang kadang menjelaskannya dalam bahasa Sunda."* (CSR-Informant Star energy 3)

> "I joined a yoghurt making training; the trainer cannot speak Sundanese language. If you use the Sundanese language, it would be much better to understand. Luckily some friends sometimes explain it in Sundanese language." (Star energy's CSR-Informant 3)

> *"Saya baru pertama kali gagal dalam program ini, nah ini pertama kalinya saya, tapi untungnya teman saya membantu menunjukkan cara yang benar tentang cara melakukannya."* (CSR-Informant Star energy 7)

> "I had failed for the first time in the program, well it is my first time, but fortunately my friend help showing the right way on how to do it." (Star energy's CSR-Informant 7)

> *"Not all participants who come to the programs had the same experiences. In our training of making this cassava chips, some participants had practised it. But the results were not very good, the texture of chip was hard. He shares his experiences and problem, and the facilitator gives us solutions on how to fix it."* (PGE CSR-Informant 6)

> "Tidak semua peserta yang datang ke program memiliki pengalaman yang sama. Dalam pelatihan kami membuat keripik singkong ini, beberapa peserta telah mempraktekkannya. Tapi hasilnya kurang bagus, tekstur chipnya keras. Dia berbagi pengalaman dan masalahnya, dan fasilitator memberi kami solusi tentang cara memperbaikinya." (PGE CSR-Informant 6)

> *"Adalah permintaan kami kepada PGE untuk memberikan kami pelatihan budidaya jamur. Karena banyak orang bilang, pasarnya bagus. Selama latihan, saya gagal tetapi beberapa teman berhasil. Jadi saya belajar darinya"* (PGE CSR-Informant 7)

> "It was our request to the PGE to provide us mushrooms cultivation training. Since many people say, the market was good. During the practice, I failed but some friends succeeded. So, I learned from him" (PGE CSR-Informant 7)

In addition to the items mentioned above, and is the most important step in the CSR acceptance process and has an effect on the target community, it will develop involvement that is not only felt as a community that has its own identity but also as something that is

owned by individual members. Even the participants themselves may define a successful program that fosters a sense of togetherness and intimacy among the participants.

> *"Alhamdulillah, saya tidak hanya senang bisa mengikuti program ini, tapi juga senang bisa melewati masa-masa sulit bersama. Kami semakin dekat satu sama lain"* (Informant CSR Star energy 4)

> "Alhamdulillah, I am not only happy to have joined this program, but it is my pleasure to be able to get through difficult times together. We are getting closer to each other" (Star Energy's CSR-Informant 4)

> *"Saya sangat senang bisa mengikuti kegiatan yang dilakukan Pertamina. Dulu saya diundang oleh teman-teman untuk mengisi waktu luang, tetapi saya merasakan banyak hal baru dan bahkan hubungan kami dengan orang lain menjadi lebih dekat sebelum mengikuti program ini. Jika saya melakukan sesuatu, jika program ini berlanjut, saya ingin mengundang ibu-ibu lain."* (PGE's CSR- Informant 2)

> "I am very happy to be able to participate in the activities carried out by Pertamina. I used to be invited by friends to fill my spare time, but I felt a lot of new things and even our relationship with other people was closer before joining this program. If I do anything, if this program continues, I want to invite other mothers." (PGE's CSR-Informant 2)

In other words, a program that is successful and achieves the level of engagement expected from participants will foster a sense of togetherness. The community needs to do this in order to make economic progress that is sustainable. Successful participants in the first phase of the program can act as change agents to ensure that it continues to be successful in the future.

A CSR program becomes close and acceptable to the target community by maximizing social capital and having the ability to recognize supportive variables. We also suggest that it will lower community expectations for transactional behavior related to program execution. According to Rudito, Famiola, and Anggahegari (2018), program participants who take part in this agenda frequently give brief introductions to the program, like receiving a reward for attending training, especially if it is hosted by a sizable company whose reputation is well-known to the general public.

If every empowerment program through CSR pays close attention to how each stage in the program relates to interpersonal reactions inside the agency, different motives for participation in the program may also be adjusted. In order to speed up anticipation and prompt reaction and prevent participants from losing focus and momentum from the program's implementation goals, it is crucial that the CSR program facilitators establish strong bonds with program participants.

## 5. Conclusions and Recommendations

This study's conclusion raises important questions about how social capital affects a community's or a group's participation in programs designed to increase their ability or financial accessibility, notably in CSR activities. Social capital might foresee "individual reasons" in the early stages of the program, especially economic motives like receiving monetary rewards, which could be eliminated. Many people are eager to participate in an activity for financial reasons, and even when the program offers them non-financial advantages like fresh perspectives and information, they are unaware of or unconcerned about this. Early in the program's preparation, it would be possible to minimize participation costs by gaining knowledge of the social interaction and relationship patterns among potential participants.

Participants in the program also benefit from social capital, which keeps them engaged even when they are still unsure of how to proceed. In other words, having strong social capital may prevent participants from quitting the program too soon. This study discovered that individuals who got along well with one another were more likely to adjust swiftly to the program's activities. The participants' intimate bonds also foster an environment where

participants may openly discuss their issues and develop understanding, which helps the program continue in the future and meet its goals.

The development of group commitment and morale for the longevity of the program is another crucial function of social capital. In order to bind individual participant commitment to mutual commitment among participants for their future well-being, social capital becomes a crucial component.

In conclusion, social capital is crucial to the success of CSR initiatives that promote community empowerment. Therefore, it is imperative that organizations planning to engage in such activities pay substantial attention to conducting in-depth analyses of social capital. The program must understand the social capital of the community it is targeting in order to succeed.

Accordingly, this study has shaped the theoretical understanding related to social capital. It provides new insight into understanding the role of social capital at the macro level of implementing an empowerment program. While practically speaking, this study provides insight into how necessary social capital is to create community engagement. Using this knowledge, companies could test whether their CSR agenda has reached a certain level of beneficial community engagement in the program according to the progress.

The limitation of this study is we use only three case sample CSR programs that are implemented basically in homogeneous cultures, namely West Java in Indonesia. Empirical testing is needed to validate whether the roles of social capital described in this study also apply to CSR practices elsewhere so that it will sharpen the prepositions we explain in this study. We firmly believe they are a factor. Local institutional nature will contribute greatly in the form of social capital and its contribution to social change from CSR agendas or other models of community empowerment programs.

In the future, we recommend that more case studies from different parts of Indonesia, as well as other countries, and may take a look at diversity and local conditions in conducting CSR programs.

**Author Contributions:** Methodology, P.A.; Formal analysis, M.F.; Writing—original draft, B.R. All authors have read and agreed to the published version of the manuscript.

**Funding:** We received funding from School of Business and Management, Bandung Institute of Technology.

**Institutional Review Board Statement:** The study was conducted in accordance with the Declaration of Helsinki, and approved by the Institutional Review Board (or Ethics Committee) of School of Business and Management, Bandung Institute of Technology (069/I1.C12/SK/PL/2020 on 10 February 2020).

**Informed Consent Statement:** We have asked each informant's consent.

**Data Availability Statement:** Any required data will be provided with a request in writing via the email of the corresponding authors listed due to privacy of informants.

**Conflicts of Interest:** The authors declare no conflict of interest.

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
