# Peer review of "Corporate Social Responsibility and Social Capital: Journey of Community Engagement toward Community Empowerment Program in Developing Country"

_sustainability, doi:10.3390/su15010466_

Round 1

Reviewer 1 Report

Dear authors,

Greetings!

After reviewing your manuscript, i would like to present the following comments:

1- Regarding the abstract, i think there are some sentences missed; it seems something to be continued after the last word "community". However, abstract should include purpose of study, data collection method, main findings and   recommendations. 

2- This study is a narration of events that took place between the concerned parties, and that the study of three cases cannot, in my opinion, reflect a study whose results can be relied upon. In fact, this study as it appears describes the behavior of the three companies with regard to community engagement.

3- Accordinglly, i can see that this study is not appropriate to be published in "Sustainability".

 Thanks and regards,

Author Response

Dear Reviewer, We thank you for your review. Below are the list of response. 

1- Regarding the abstract, I believe some sentences are missing; after the word "community," it seems as though something should have been continued. However, the study's objectives, data collection strategy, key findings, and suggestions ought to be included in the abstract.

-We replace the word "community" with the word "togetherness." We believe the inclusion of the word "community" in the phrase "sense of community" may be confusing.

2- This study is a narration of events that took place between the concerned parties, and that the study of three cases cannot, in my opinion, reflect a study whose results can be relied upon.

- In fact, this study as it appears describes the behavior of the three companies with regard to community engagement. The object study the behaviour of community not the behavior of corporation. We explain more detail on how the data collected may be beneficial for the community participating in three CSR program. The three CSR program in this stage just support in regards to identifying suitable informants for the study.

3- Accordingly, I can see that this study is not appropriate to be published in "Sustainability" 

- Thank you for your suggestion, however we do think that this paper is original and provide another point of view which are align with the topic of this journal

Reviewer 2 Report

This article discussed the Corporate Social Responsibility and Social Capital: Journey of Community Engagement toward Community Empowerment Program in Developing Country. I have the following comments for improvement:

1. Abstract needs to be rewritten and emphasized on the key findings of this research.

2. Introduction section is poorly written. I recommend authors to discussed the key aspects of corporate social responsibility, including the objective of this study and scope of this research.

3. Literature review section can be further improved through citing the up-to-date literature. Following articles can consider:

https://doi.org/10.3390/su132212790

https://doi.org/10.1108/MEQ-08-2020-0178

4. Methodology section is ok

5. Results and analysis are fine

6. Conclusion section can be further improve through discussing the practical implications, theoretical contribution, future research avenue and limitations of current study.

Author Response

Dear Reviewer, We thank you for your review. Below are the list of response. 

  1. Abstract needs to be rewritten and emphasized on the key findings of this research: We had rewritten the abstract to meet the suggestion given by reviewer
  2. Introduction section is poorly written. I recommend authors to discussed the key aspects of corporate social responsibility, including the objective of this study and scope of this research: In the introduction we have include the research question to make the objective of the research is clear
  3. Literature review section can be further improved through citing the up-to-date literature. We had added more update relevant references

Following articles can consider:

https://doi.org/10.3390/su132212790

https://doi.org/10.1108/MEQ-08-2020-0178

We have included the literature

  1. Methodology section is ok: Thank you
  2. Results and analysis are fine: Thank you
  3. Conclusion section can be further improve through discussing the practical implications, theoretical contribution, future research avenue and limitations of current study:  We have add the implication of the study to theoretical and practical implementation of the study. We have also put the future potential research.

Reviewer 3 Report

This is a very relevant, interesting and important topic. Further research can definitely be developed following same. 

The referencing can be further improved.

Author Response

This is a very relevant, interesting and important topic. Further research can definitely be developed following same. The referencing can be further improved.

Thank you for the comment

Reviewer 4 Report

This is paper is well poised to make a contribution to the field. There are some matters for clarification/elaboration. These are as follows:

1. Introduction-This can be better suited and supported by existing research in the field.  Otherwise, this introduction does not appear to be grounded in the literature.

2. Research questions/objectives can be better linked to the role of social capital. 

3. Strengthen analysis-to better link peer invitation, involvement, and connections/networks to the social capital. 

4.Discussion-should be better suited within the literature with some treatment of the implications for theory and practice.  

Author Response

Thank you very much for the suggestion. 

  1. Introduction-This can be better suited and supported by existing research in the field. Otherwise, this introduction does not appear to be grounded in the literature: We have revise the introduction section
  2. Research questions/objectives can be better linked to the role of social capital. We added this in the research question: we have include the research question in introduction section
  3. Strengthen analysis-to better link peer invitation, involvement, and connections/networks to the social capital: This information has now be included in discussion

4.Discussion-should be better suited within the literature with some treatment of the implications for theory and practice: we had added and linked the finding and with literature in particular discussion

Round 2

Reviewer 1 Report

Dear authors,

Thank you so much for your efforts in improving the research. However, few comments to be considered:

1- In the abstract, it is very necessary to identify the methodology "case studies" of your study.

2- In the findings and discussion, since you have already presented litrature review, it is better to support your results with some similar results from other studies.

3- Add recommendations in the conclusion section. Also, mention to the same in the abstract.

All the best.

Author Response

Dear Reviewer, thank you for the suggestion. 

1- In the abstract, it is very necessary to identify the methodology "case studies" of your study.

Response to Reviewer: The word case studies has been inserted in the abstract

2- In the findings and discussion, since you have already presented literature review, it is better to support your results with some similar results from other studies.

Response to Reviewer: The Explanation has been incorporated 

3- Add recommendations in the conclusion section. Also, mention to the same in the abstract.

Response to Reviewer: The Explanation has been incorporated 

Reviewer 2 Report

Thanks for the revised manuscript submission. However, I noticed that the authors did not submit "Author response file"

It is strongly recommended to submit your "author response sheet" and reply to each comments properly. Also, I noticed that authors did not revised the article properly too.

Author Response

Dear Reviewer,

Thank you for notifying us. We apologize for not uploading the Author Response sheet. Both comments and suggestion has now been incorporated and attached in the response sheet.

Thank you

Round 3

Reviewer 2 Report

This article is ready for publication.